# Long-Term Effects of Tolvaptan in Autosomal Dominant Polycystic Kidney Disease: Predictors of Treatment Response and Safety over 6 Years of Continuous Therapy

**DOI:** 10.3390/ijms25042088

**Published:** 2024-02-08

**Authors:** Mai Yamazaki, Haruna Kawano, Miho Miyoshi, Tomoki Kimura, Keiji Takahashi, Satoru Muto, Shigeo Horie

**Affiliations:** 1Department of Urology, Graduate School of Medicine, Juntendo University, Tokyo 113-8431, Japan; m.yamazaki.fh@juntendo.ac.jp (M.Y.);; 2Department of Advanced Informatics for Genetic Diseases, Graduate School of Medicine, Juntendo University, Tokyo 113-8431, Japan; 3Department of Urology, Juntendo University Nerima Hospital, Tokyo 177-8521, Japan

**Keywords:** ADPKD, tolvaptan, BMI, biomarkers, family history

## Abstract

Tolvaptan, an oral vasopressin V2 receptor antagonist, reduces renal volume expansion and loss of renal function in patients with autosomal dominant polycystic kidney disease (ADPKD). Data for predictive factors indicating patients more likely to benefit from long-term tolvaptan are lacking. Data were retrospectively collected from 55 patients on tolvaptan for 6 years. Changes in renal function, progression of renal dysfunction (estimated glomerular filtration rate [eGFR], 1-year change in eGFR [ΔeGFR/year]), and renal volume (total kidney volume [TKV], percentage 1-year change in TKV [ΔTKV%/year]) were evaluated at 3-years pre-tolvaptan, at baseline, and at 6 years. In 76.4% of patients, ΔeGFR/year improved at 6 years. The average 6-year ΔeGFR/year (range) minus baseline ΔeGFR/year: 3.024 (−8.77–20.58 mL/min/1.73 m^2^). The increase in TKV was reduced for the first 3 years. A higher BMI was associated with less of an improvement in ΔeGFR (*p* = 0.027), and family history was associated with more of an improvement in ΔeGFR (*p* = 0.044). Hypernatremia was generally mild; 3 patients had moderate-to-severe hyponatremia due to prolonged, excessive water intake in response to water diuresis—a side effect of tolvaptan. Family history of ADPKD and baseline BMI were contributing factors for ΔeGFR/year improvement on tolvaptan. Hyponatremia should be monitored with long-term tolvaptan administration.

## 1. Introduction

In autosomal dominant polycystic kidney disease (ADPKD), cysts develop in both kidneys; these grow and enlarge over decades, forming numerous cysts and causing impaired kidney function. The global incidence of ADPKD is 1 in 1000–4000 people, with *PKD1* and *PKD2* as the known causative genes [1]. The penetrance of ADPKD is 100%, and the disease is often familial, but de novo mutations occur in 5–20% of cases [2]. Decline in renal function occurs at a faster rate in patients with *PKD1* mutations than in those with *PKD2* mutations, and the age at which end-stage renal disease (ESRD) occurs can vary by 20 years [3,4]. 

While ADPKD causes various systemic complications in organs other than the kidneys, including liver cysts, pancreatic cysts, cerebral aneurysms, and valvular heart disease, the most clinically relevant manifestation is the loss of renal function. Tolvaptan is an oral vasopressin V2 receptor antagonist that is effective in reducing volume in the enlarged kidney and improving declining kidney function [5,6,7,8,9,10,11,12].

In TEMPO3:4, an international phase III clinical trial, tolvaptan showed efficacy in reducing renal volume growth and renal function decline in patients with ADPKD (18–50 years), with a total kidney volume (TKV) > 750 mL, and an annual TKV growth rate of ≥5%. Estimated glomerular filtration rate (eGFR) decline was reduced by 35%/year (per 1.73 m^2^), improving from −3.70 to −2.72 mL/min [5,12]. Following the TEMPO3:4 trial, a second phase III study (REPRISE) demonstrated that tolvaptan was also effective in patients with more advanced disease: eGFR 25–65 mL/min/1.73 m^2^ (56–65 year-old patients), and eGFR 25–44 mL/min/1.73 m^2^ (56–65 year-old patients) [6,11,13].

Based on the data from these two randomized controlled trials (RCTs), tolvaptan is now widely used globally for the treatment of ADPKD. However, one drawback of this treatment is the requirement for patients to drink large amounts of water, and the resulting water diuresis causes a significant increase in urine output [6]. For instance, at the maximum tolvaptan dose of 120 mg/day, an average urine volume of 3–7 L/day has been reported in patients with stage 1-to-stage 2 (G1 to G2) chronic kidney disease (CKD), and patients require oral fluid rehydration to compensate for this urine volume [14]. This water diuresis, known as aquaresis, reduces tolvaptan tolerability, and is also a barrier to initiating treatment, as patients are expected to remain on tolvaptan until they reach ESRD, which means a large daily water intake over many years. The risks and benefits of tolvaptan treatment should, therefore, be fully considered before initiating treatment. 

The Mayo Classification identifies high-risk patients with ADPKD into five categories (1A, 1B, 1C, 1D, and 1E), and it is the standard index for tolvaptan treatment. The consensus recommends that tolvaptan treatment is used for patients with a Mayo Classification of 1C or higher, based on the risk of disease progression and the efficacy of tolvaptan to prevent renal function decline [15,16]. There is, however, a lack of research into the predictive factors that indicate which patients are more likely to benefit from tolvaptan treatment in the long term.

Considering this unmet need, the aim of this study was to focus on patients who had been treated with tolvaptan over a number of years, to investigate potential predictors of treatment response, and to assess efficacy and safety.

## 2. Results

### 2.1. Patient Baseline Characteristics

A total of 130 patients with ADPKD visited Juntendo University Hospital up until November 2023. Of these, patients who had started tolvaptan before November 2017 were selected. A total of 55 patients who had continuously received tolvaptan for more than 6 years (72 months) were included in this study (Figure 1).

Patient characteristics are shown in Table 1. Of the 55 patients included in the study, the majority were men (72.7%), and the average body mass index (BMI) (range) was 23.2 (17.7–32.5) kg. The average age (range) at treatment initiation was 46.7 (24–72) years. The majority of patients (80%) had a family history of ADPKD. The Mayo Classification categories at baseline included the following: Class 1A, *n* = 0; Class 1B, *n* = 11 (20.0%); Class 1C, *n* = 17 (30.9%); Class 1D, *n* = 19 (34.5%); Class 1E, *n* = 8 (14.5%). Of the 55 study patients, 39 patients had previously received genetic testing [17]: *PKD1* truncated genetic mutations, *n* = 22 (40.0%); *PKD1* non-truncated, *n* = 7 (12.7%); *PKD2* truncated, *n* = 7 (12.7%); *PKD2* non-truncated, *n* = 4 (7.2%) [17], (Table 1).

ADPKD, autosomal dominant polycystic kidney disease; BMI, body mass index; IQR, interquartile range; PKD, polycystic kidney disease.

### 2.2. Tolvaptan Dosing

The starting dose of tolvaptan was 60 mg/day, gradually increasing to a maximum dose of 120 mg/day (a lower dose was used if the eGFR was less than 30 mL/min/1.73 m^2^). Dose reductions were considered based on patient tolerability.

The median starting dose for the 55 patients in the study was 60 mg/day (*n* = 34 patients). A total of 14 patients started on 30 mg/day; 5 patients on 45 mg/day; and 1 patient each on 7.5 mg/day, and 15 mg/day. The median daily dose over 72 months was 90 mg/day, and 26 patients (47.3%) received the maximum 120 mg/day dose.

### 2.3. Effect of Tolvaptan on Renal Function

The average eGFR (range) at baseline was 55.5 (24.6–112.7) mL/min/1.73 m^2^. After 6 years of tolvaptan treatment, the average eGFR (range) was 36.6 (6.2–81.9) mL/min/1.73 m^2^. The decrease in eGFR over 1 year (ΔeGFR/year) at baseline was: −5.74 (−22–6.4) mL/min/1.73 m^2^; 6 years −2.72 (−7.98–1.98) mL/min/1.73 m^2^, Table 2. Waterfall plots of ΔeGFR/year are shown in Figure 2.

Compared to baseline, ΔeGFR at 6 years improved in 76.4% of patients, and the mean (range) change was 3.024 (−8.772–0.58) mL/min/1.73 m^2^.

When ΔeGFR/year was compared at 6 years to baseline, the mean (range) change was 3.024 (−8.772–0.58) mL/min/1.73 m^2^. After 6 years of tolvaptan treatment, ΔeGFR/year improved (6-year ΔeGFR/year minus baseline ΔeGFR/year > 0) in 76.4% of patients: mean (range), 4.95 (0.14–20.6) mL/min/1.73 m^2^; in 23.6% of patients, ΔeGFR/year did not improve: −3.19 (−8.7748–−0.12) mL/min/1.73 m^2^. At baseline, ΔeGFR/year was significantly better in the non-improved group than in the improved group (−0.183 mL/min/1.73 m^2^, *p* < 0.001). A total of 5 patients had a baseline ΔeGFR/year of ≥0 (ΔeGFR unchanged or improving) and were all in the ΔeGFR/year worsening group. 

### 2.4. Predicted 6-Year eGFR and Actual 6-Year eGFR

Predicted 6-year eGFR was calculated from baseline eGFR with the Mayo classification, which predicts the rate of decline in renal function based on baseline HtTKV and age. 

The percentage of patients with a better-than-predicted eGFR minus actual 6-year eGFR > 0 was 50.9%, with a mean (range) of 11.0 (0.58–41.0) mL/min/1.73 m^2^. The percentage of patients with a worse-than-predicted eGFR minus actual 6-year eGFR > 0 was 49.1%, with a mean (range) of 11.1 (−27.9–0.22) mL/min/1.73 m^2^. 

Factors related to the ‘predicted eGFR minus actual eGFR’ were examined using Pearson’s correlation coefficient, and a negative correlation was found with baseline eGFR, with lower original eGFR values tending to be significantly lower than predicted eGFR values at 6 years (*p* = 0.0219).

Previous studies have shown that the decrease in ΔeGFR is slower in female patients than in males [15]. A study was conducted to see if there was a difference in ΔeGFR between men and women, but the results of the *t*-test analysis comparing ΔeGFR in men and women showed no significant difference (*p* = 0.133).

### 2.5. Effects of Tolvaptan on TKV

The mean HtTKV (range) at baseline was 1146.6 (412.0–3003.1) mL, and at 6 years was 1390.9 (406.1–4067.7) mL. The waterfall plot of HtTKV is shown in Figure 3.

The mean percentage increase in TKV at 1 year (ΔTKV %) at baseline was 7.65% (−14–26.1%). In the REPREASE study, ΔTKV was known to improve at 3 years and then worsen due to re-increase [6,11,13]. In this study, the 3-year and 6-year ΔTKV were 7.3% (−24.1–38.8%) and 21.9% (−32.0–99.1%), respectively, which is similar to those previously reported. ΔTKV was compared using the Friedman test between the three groups: baseline, 3 years, and 6 years. ΔTKV showed a significant increase in change between baseline and up to 6 years (*p* < 0.05), no significant difference in change between baseline and 3 years, but a significant increase in change from 3 years to 6 years (*p* < 0.001) (Figure 4).

A *t*-test was conducted to examine the effect of gender on ΔTKV%, but no statistically significant difference was observed between the male and female patients (*p* = 0.245).

The relationship of tolvaptan with TKV and renal function was examined. Patients were divided into ΔTKV improved/unimproved groups based on whether their 6-year ΔTKV improved compared to baseline, and then similarly divided into ΔeGFR improved/unimproved groups by comparing their 6-year ΔeGFR/year and baseline ΔeGFR/year. These data, analyzed using the Fisher test, showed that there were no significant differences between improvement in ΔTKV and improvement in ΔeGFR at 6 years (*p* > 0.05).

### 2.6. Analysis of Predictors of Treatment Efficacy of Tolvaptan

A multivariate analysis of factors associated with the therapeutic effect on renal function was conducted. Factors related to ΔeGFR improvement (calculated as 6-year ΔeGFR/year minus baseline ΔeGFR/year) were selected from previous reports [5,6,7,8,9,10,11,12,13,14] and analyzed for correlation. These included the following: Mayo classification (1C or higher), tolvaptan daily dose (60 mg or higher), baseline BMI, presence of family history, and hypertension (Table 3). Baseline BMI and family history were found to have a statistically significant effect on treatment response (*p* = 0.027, *p* = 0.044, respectively) (Table 3). A higher BMI was associated with less of an improvement in ΔeGFR, and family history was associated with more of an improvement in ΔeGFR. There was no significant correlation found between tolvaptan daily doses and improvement in ΔeGFR. A stratified analysis was performed by tolvaptan dose, with mean daily doses of less than 60 mg, 60-to-90 mg, and 90 mg or more being analyzed using the Kruskal–Wallis test. However, no correlation was found between the dose and the degree of improvement in ΔeGFR in all strata (*p* = 0.09).

The BMI examination at 6 years showed a mean BMI of 23.11 (range: 16.43–33.4) and a mean difference of −0.118 (−9.27–7.48) compared to the baseline. Although Spearman’s rank correlation coefficient was used to analyze the change in BMI and improvement in ΔeGFR, no significant correlation was found (*p* = 0.328).

### 2.7. Safety and Long-Term Tolerability of Tolvaptan

Blood samples were taken monthly for six years during the patients’ treatment with tolvaptan. Liver enzymes (AST, ALT, or γ-GTP) exceeded 1.5-times the reference value at least once in 9 patients (16.3%) and exceeded 3-times the reference value at least once in 4 patients (7.2%).

The doses of tolvaptan at the onset of liver dysfunction were 15 mg (1 patient), 30 mg (3 patients), 40 mg (1 patient), 45 mg (1 patient), 60 mg (2 patients), and 120 mg (2 patients). Of these patients, none discontinued tolvaptan treatment, and 4 patients had a dose reduction. For the patient with the most severe hepatic impairment, the liver enzyme levels at the time of tolvaptan induction were: AST 31 IU/L, ALT 166 IU/L, and γ-GPT 394 IU/L. At 12 months of treatment (90 mg tolvaptan), their levels were: AST 28 31 IU/L, ALT 158 31 IU/L, and γ-GPT: 431 31 IU/L. No patients progressed to the parameters stated in Hy’s law (drug-induced liver injury resulting in ALT > 3-times the ULN, total bilirubin > 2-times the ULN) [5]. In 25% of patients where a dose reduction had occurred, the tolvaptan dose was increased to the original dose or higher.

Hypernatremia (sodium [Na] ≥ 146 mmol/L) occurred in 17 patients (31%) over 6 years (median Na 146 mmol/L), with most instances remaining mild; the highest value was 147 mmol/L. These patients were encouraged to drink water, and tolvaptan treatment was not discontinued, nor was the dose reduced. Hyponatremia (Na ≤ 134 mmo/L) occurred in 7 patients (12.7%) during treatment (median Na 130 mmol/L); no adverse effects were evident for most patients. Out of these 7 patients, there were 3 who had moderate or severe hyponatremia, 2 with moderate (Na 126 mmol/L) hyponatremia, and another 1 with severe hyponatremia (Na 117 mmol/L). In a severe case, this occurred at month 64 (tolvaptan 120 mg), with a fluid intake of 6–7 L/day. This patient had chronic moderate hyponatremia with Na 125–130 mmol/L and Na 117 mmol/L, headache and malaise, but no severe symptoms. The patient recovered spontaneously after limiting fluid intake to about 5 L/day and increasing salt intake; tolvaptan treatment was continued (Appendix A).

## 3. Discussion

The aim of tolvaptan in clinical practice is to extend the time to ESRD in patients with CKD. The average age at which untreated CKD patients reach ESRD is generally 60 years [18,19,20,21,22]. In the TEMPO3:4 study, tolvaptan was reported to improve yearly eGFR decline (ΔeGFR) by 1 mL/min/1.73 m^2^ compared to placebo (placebo −3.70, tolvaptan −2.72 mL/min/1.73 m^2^) in patients with ADPKD [5,12]. Tolvaptan in the REPRISE trial also improved ΔeGFR by 1.27 mL/min/1.73 m^2^ (placebo −3.61, tolvaptan −2.34 mL/min/1.73 m^2^) [6,13]. In this study, the change in ΔeGFR was 3.024 mL/min/1.73 m^2^, improving ΔeGFR by about 40%. However, 13 patients (23.6%) experienced a worsening of ΔeGFR/year during the 6-year period. These patients had a significantly better baseline ΔeGFR/year (−0.183 mL/min/1.73 m^2^) compared to the patients with an improved ΔeGFR/year (*p* < 0.001). All 5 patients with a baseline ΔeGFR/year ≥ 0 (ΔeGFR unchanged or improving) had a worsening ΔeGFR/year during the 6-year period. These data raise the question as to whether patients with a large TKV, currently considered for tolvaptan treatment, but with no decline in renal function, should delay the start tolvaptan treatment. A subgroup analysis of TEMPO 3:4 found that the effect of tolvaptan on eGFR was limited and not significant in patients with a normal eGFR (>90 mL/min/1.73 m^2^) [5]. In patients with ADPKD, eGFR is normal due to nephron compensation, until it begins to decrease during disease progression [4]. Treatment decisions in patients with a high HtTKV who are considered for tolvaptan induction should take into account the decline in renal function after nephron compensation is no longer effective [20,23,24].

For patients with a baseline ΔeGFR/year of ≥0 (ΔeGFR unchanged or improving), it is likely that tolvaptan treatment was initiated during this ‘nephron compensation period’ and that patients entered a period of renal decline during the 6 years of treatment, resulting in a pre- and post-ΔeGFR/year comparison that may have given the impression of no treatment effect [20,23,24]. Observing the change in ΔeGFR/year alone suggests that these patients were started on tolvaptan too early. However, it is well known that treatment benefit, in terms of preserving renal function, is enhanced if tolvaptan is introduced before disease progresses. This is particularly the case for younger patients who may have rapidly progressing ADPKD despite normal eGFR, and who need to be treated with caution [19]. Therefore, in clinical practice, as well as evaluating baseline ΔeGFR/year, a combination of other factors, including genetic mutations, family history, and comorbidities need to be considered before the decision to start tolvaptan is made. Regular data collection on renal function trends, even in the presence of normal renal function, makes it easier to detect a decline in ΔeGFR/year, and greatly facilitates the decision to initiate tolvaptan [25]. Regular monitoring of renal function should be performed well-before tolvaptan induction is considered. The European Renal Association (ERA) Working Group states that ΔeGFR ≤ 3 mL/min/1.73 m^2^ is defined as rapid disease progression [19], and that ΔeGFR must be reliable and based on at least five eGFR measurements over at least 4 years. This study used ΔeGFR calculated from renal function measurements within 3 years, which may be insufficient in terms of duration and frequency, and a potential limiting factor in this study.

Factors related to ΔeGFR improvement were analyzed by multiple regression analysis. Baseline BMI and family history were found to significantly influence treatment effect (*p* = 0.027 and *p* = 0.044, respectively). Higher BMI correlated with less improvement in ΔeGFR, and better correlation with family history was associated with greater improvement in ΔeGFR (Table 3). There was also a trend towards a better ΔeGFR with an average daily dose of tolvaptan of 60 gm/day or more, although there was no predominant difference.

ADPKD is mainly familial, but de novo mutations are known to occur with a frequency of 5–20% [2], although the site of predilection for de novo mutations is unclear. The possibility of having a *PKD1* truncating mutation is increased if a blood relative who has developed ESRD before the age of 60 years is present [26]. In this study, 80% of patients had a family history of ADPKD. These patients tended to have a higher baseline TKV and ΔTKV, but better baseline eGFR and ΔeGFR, than patients without a family history of ADPKD, although these differences were not statistically significant (Appendix A). While having a blood relative with severe ADPKD may encourage a patient’s referral and access to healthcare at a time when renal function is still preserved and tolvaptan treatment is considered due to increased renal volume [27], for patients with no family history of ADPKD, access to early medical care remains an issue.

Obesity is a risk factor for the development or progression of CKD [28,29]. Studies in mouse models of ADPKD have shown defective glucose metabolism and metabolic reprogramming; however, mild-to-moderate dietary restriction can slow disease progression [30,31,32,33]. Nowak et al. (2018) analyzed the association between ADPKD and obesity in 441 non-diabetic patients with an eGFR > 60 mL/min/1.73 m^2^ (PKD-HALT Study A). These patients were divided into three groups according to BMI: normal weight (18.5–24.9 kg/m^2^), overweight (25.0–29.9 kg/m^2^), and obese (≥30 kg/m^2^). The results showed that overweight patients, especially obese patients, showed a greater yearly change in TKV, and a greater reduction in ΔeGFR [34,35]. A total of 1312 patients in the TEMPO 3:4 study were also analyzed to determine the association between baseline BMI and change in TKV over a 3-year study period. Findings from TEMPO 3:4 showed that the higher the BMI, the greater the yearly change in TKV. The decrease in ΔeGFR did not change with respect to BMI [36]. This study showed a difference in the decline of ΔeGFR with respect to BMI, but not in the annual rate of change for TKV. While these findings vary from the reported literature, possibly due to the small number of patients in this real-life, retrospective study, these data are nonetheless informative for future clinical practice, as higher BMI is associated with reduced treatment efficacy with tolvaptan.

Exercise and dietary therapy are useful in the treatment of obesity and have recently received increased attention as a treatment for ADPKD. Calorific restriction inhibits the mammalian target of the rapamycin (mTOR) signaling pathway by reducing the adenosine triphosphate/adenosine monophosphate (ATP/AMP) ratio and activating AMP-activated protein kinase (AMPK). The mTOR pathway is aberrantly activated in ADPKD kidney cysts [1,2]. In a mouse model of PKD, a 23% reduction in food intake was shown to be more potent in inducing mTOR inhibition than pharmacological mTOR inhibition [33]. Restricting calories in patients with ADPKD who are overweight has also been shown to have a beneficial effect on liver cysts [37]. Whether the differential effects of BMI on ΔeGFR in this study are due to the adverse effects of being overweight or obese, or the possible additional mTOR inhibitory effects of a normal-to-lean body mass and a low calorie diet, is for future research.

Considering the long-term tolerability of tolvaptan in this study, liver dysfunction greater than 1.5 times the reference value occurred more frequently than previously reported [5,6,38]. However, all patients continued treatment, either by maintaining their dose or with a dose reduction. Monthly liver function monitoring and dose reduction contributed to the safe continuation of tolvaptan. Hypernatremia, a side effect of concern with tolvaptan, was generally mild. Hyponatremia was mild in terms of subjective symptoms, but some patients experienced severe hyponatremia [39]. Low sodium levels should be noted in long-term tolvaptan use, as salt reduction and excessive water consumption may be habitual [40,41,42].

## 4. Materials and Methods

### 4.1. Study Population and Parameters

In this retrospective study, clinical data from medical records were collected from sequential patients diagnosed with ADPKD attending Juntendo University Hospital, Tokyo from November 2017 to November 2023. Data for kidney function evaluation were collected pre-tolvaptan (36 months prior to tolvaptan start), at baseline (before starting tolvaptan), 1 month after the start of tolvaptan, and every 3 months thereafter up to 6 years (72 months) of treatment. Data for safety were collected every 1 month thereafter up to 6 years (72 months) of treatment. Imaging studies (standardized kidney computed tomography [CT] scans) were collected pre-tolvaptan (36, 24, and 12 months prior to tolvaptan start), at baseline, and every 12 months thereafter up to 6 years (72 months) of treatment.

### 4.2. Data Collection and Calculations

Renal function and progression of renal dysfunction were assessed using eGFR, and 1-year change in eGFR (ΔeGFR/year). Estimated GFR was calculated using the equation (Japanese population): eGFR = 194 × (serum creatinine)^−1.094^ × (age)^−0.287^ × (0.739 for women) [43]. In addition, ΔeGFR/year was calculated from an approximation curve based on eGFR values measured over 3 years prior to starting tolvaptan.

Total kidney volume (TKV) was assessed and measured using CT, and measurements were performed by a single urologist assessor to avoid potential bias and irregularities. Total kidney volume was estimated using the ellipsoid rotational volume method: (π/6 × long diameter × [small diameter]^2^). Height-adjusted TKV (HtTKV) was used in this study as it has been shown to correlate with renal function without the need to consider differences in men and women [3,4]. An approximation curve was constructed from TKV values measured over time from 3 years prior to tolvaptan initiation, and percentage increase in TKV/year (ΔTKV/year) was calculated.

Using the age of the patient and the calculated HtTKV, the imaging classification of ADPKD was obtained from the web-based Mayo classification model for typical ADPKD patients (Class 1) [15]. There were no atypical (Mayo Class 2) ADPKD patients in the study population.

### 4.3. Statistical Analyses

Continuous variables are reported as average deviation. Categorical variables are reported as percentages (%) unless otherwise stated. Analyses were conducted to determine whether changes in renal function and TKV, before and after tolvaptan administration, as well as the magnitude of change, were associated. Data were evaluated using the Student’s *t*-test, Pearson’s correlation coefficient, Friedman test, Fisher test, Kruskal–Wallis test, Spearman’s rank correlation coefficient test, and Mann–Whitney U test, as appropriate. Multiple regression analysis and logistic regression analysis were performed to determine factors related to outcomes. All statistical analyses were performed using EZR, version 1.54 (Saitama Medical Center, Jichi Medical University, Saitama, Japan) [44], and statistical significance was defined as *p* < 0.05.

## 5. Conclusions

With regular blood monitoring, long-term tolvaptan was well tolerated in patients with ADPKD who had been taking tolvaptan for 6 years. Family history and baseline BMI were found to be factors that contributed to the improvement in ΔeGFR/year with tolvaptan, suggesting that for patients with a high BMI, efforts to encourage weight loss may lead to a beneficial therapeutic effect. Physicians need to be vigilant for the occurrence of hyponatremia.

## Figures and Tables

**Figure 1 ijms-25-02088-f001:**
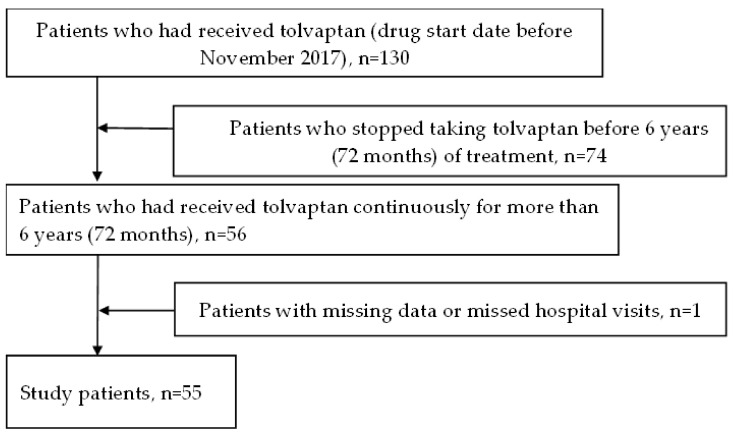
Flowchart of patients who continuously received tolvaptan for more than 6 years (72 months); drug start date before November 2017.

**Figure 2 ijms-25-02088-f002:**
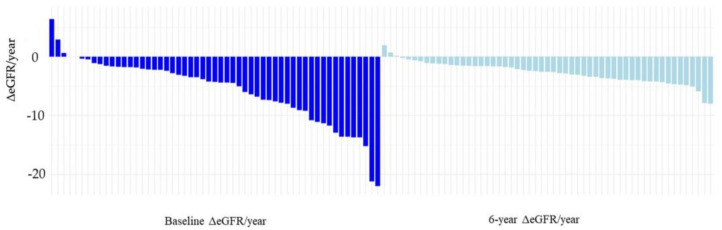
Waterfall plot: change (decrease) in eGFR over 1 year (ΔeGFR/year) before treatment with tolvaptan (baseline) and after treatment with tolvaptan (6 years).

**Figure 3 ijms-25-02088-f003:**
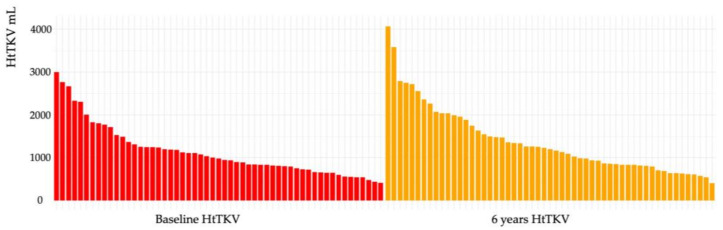
Waterfall plot: HtTKV before treatment with tolvaptan (baseline) and after treatment with tolvaptan (6 years).

**Figure 4 ijms-25-02088-f004:**
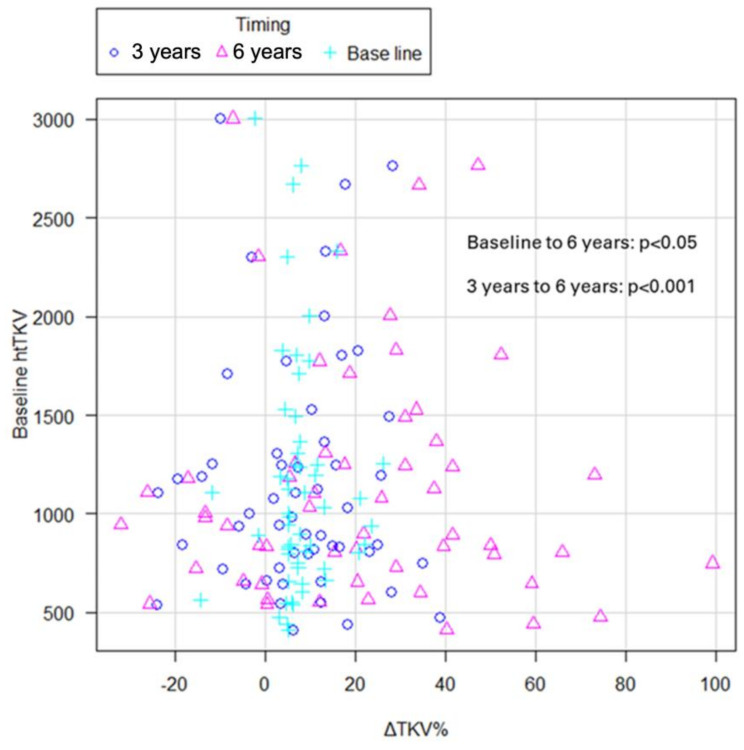
Scatter dot plot of ΔTKV% at baseline, 3 years, and 6 years with HtTKV. ΔTKV% showed a significant increase in change between both baseline to 6 years (*p* < 0.05) and from 3 to 6 years (*p* < 0.001). HtTKV, height-adjusted total kidney volume; ΔTKV%, percentage increase in TKV.

**Table 1 ijms-25-02088-t001:** Patient characteristics.

Patient Characteristics		
Patients	*n* (%)	55 (100)
Age	Average (IQR)	46.69 (24–72)
Sex		
Female	*n* (%)	15 (27.3)
Male	*n* (%)	40 (72.7)
Height, m	average (IQR)	1.69 (1.52–1.89)
BMI, kg/m^2^	average (IQR)	23.2 (17.7–32.5)
History of cerebral hemorrhage	*n* (%)	4 (7.2)
Complications of liver cysts	*n* (%)	51 (92.7)
History of heart valve disease	*n* (%)	14 (25.4)
Family history of ADPKD	*n* (%)	44 (80)
Genetic diagnosis		
unknown or untested	*n* (%)	15 (27.2)
*PKD1* Truncated	*n* (%)	22 (40)
*PKD1* Non-Truncated	*n* (%)	7 (12.7)
*PKD2* Truncated	*n* (%)	7 (12.7)
*PKD2* Non-Truncated	*n* (%)	4 (7.2)
Mayo subclass		
Class 1A	*n* (%)	0 (0)
Class 1B	*n* (%)	11 (20)
Class 1C	*n* (%)	17 (30.9)
Class 1D	*n* (%)	19 (34.5)
Class 1E	*n* (%)	8 (14.5)

**Table 2 ijms-25-02088-t002:** Patient parameters and renal function at baseline and 6 years (72 months) after initiation of tolvaptan treatment.

Patient Parameters		Baseline	After 6 Years (72 Months)
Comorbidities			
Hypertension ^a^	*n* (%)	47 (85.4)	49 (89.0)
Hyperuricemia ^b^	*n* (%)	17 (30.9)	29 (59.7)
Hyperlipidemia ^c^	*n* (%)	18 (32.7)	21 (38.1)
Cerebral aneurysm	*n* (%)	12 (21.8)	13 (23.6)
Malignant tumor	*n* (%)	5 (9.0)	8 (14.5)
Urologic ^d^ event	*n* (%)	3 (5.4)	3 (5.4)
eGFR, mL/min/1.73 m^2^	average (IQR)	55.5 (24.6–112.7)	36.58 (6.2–81.9)
ΔeGFR/year, mL/min/1.73 m^2^	average (IQR)	−4.29(−22–6.4)	−2.72 (−7.9–1.9)
TKV, mL	average (IQR)	1935.49 (710.7–5255.5)	2349.48 (682.2–6508.3)
HtTKV, mL/m	average (IQR)	1146.61 (412–3003.1)	1390.94 (406.0–4067.6)
ΔTKV%/year	average (IQR)	7.64 (−14.2–26.1)	20.74 (−32.0–74.3)

Hypertension ^a^ is defined as systolic blood pressure ≥140 mmHg, diastolic blood pressure ≥90 mmHg, or the use of antihypertensive agents. Hyperuricemia ^b^ is defined as serum uric acid ≥7.0 mg/dL, or use of uric acid lowering drugs. Hyperlipidemia ^c^ is defined as serum triglyceride ≥150 mg/dL, or use of lipids lowering drugs. Urologic ^d^ event is defined as the development of gross hematuria, ureteral stone attacks, and urinary infection.

**Table 3 ijms-25-02088-t003:** Multiple regression analysis of predictors of tolvaptan treatment response.

	Odds Ratio	Lower 95% CI	Upper 95% CI	*p*-Value	Vif
(Intercept)	13.643	0.862	26.424	0.0370	−
Mayo classification ^a^	1.939	−1.529	5.407	0.2670	1.055886
Tolvaptan ^b^ mg/day	2.948	−0.127	6.022	0.0600	1.069827
Baseline BMI	−0.570	−1.073	−0.067	0.0271 *	1.016598
Family History	−3.607	−7.118	−0.095	0.0444 *	1.082779
Hypertension ^c^	2.196	−1.745	6.138	0.2680	1.059599
			Adjusted R^2^:0.81		* = *p* < 0.05

Baseline BMI and family history had statistically significant effects on the treatment effect on ΔeGFR (*p* = 0.027, *p* = 0.044, respectively). Mayo classification ^a^: 1A, 1B/1C–1E, tolvaptan ^b^ daily doses were classified as less than 60 mg and more than 60 mg on average over 72 months, hypertension ^c^ is defined as systolic blood pressure ≥140 mmHg, diastolic blood pressure ≥90 mmHg, or use of antihypertensive agents.

## Data Availability

Anonymized data derived from this study may be provided by the corresponding author upon reasonable request.

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
