# Peer review of "Long-Term Effects of Tolvaptan in Autosomal Dominant Polycystic Kidney Disease: Predictors of Treatment Response and Safety over 6 Years of Continuous Therapy"

_ijms, 2024, doi:10.3390/ijms25042088_

Round 1

Reviewer 1 Report

Comments and Suggestions for Authors

In this manuscript, authors have conducted the effect and safety of tolvaptan treatment over a period of 6 years on 55 recruited patients. I have following concerns.

Authors should do a gender-based analysis on long-term tolvaptan effects. In the past, ΔeGFR is shown to have different outcomes in female patients than males.

Section 2.5: Authors should present the data for change in ΔTKV for all 3 groups as a scatter dot plot with their significance.

Lines 154-157: “ΔTKV showed a significant increase.... at 3 years and 6 years (p<0.001).” Authors should recheck. Line 156: “increase in change both at baseline and 6 years” seems to be repeated from line 154.

Section 2.6: Authors have analyzed multiple factors in association with therapeutic effect of tolvaptan. However, the significance of the association should be considered after correction for multiple testing (example. Bonferroni correction) especially when various factors are analyzed simultaneously.

Lines 176-177: “In addition, ΔeGFR tended.... doses of 60 gm/day or higher” implicates statistical significance between improved ΔeGFR and tolvaptan dose which is not as per p-value of 0.06 shown in table 3.

Comments on the Quality of English Language

Minor editing of English language required.

Author Response

(1) Authors should conduct a gender-based analysis on long-term tolvaptan effects. In the past, ΔeGFR has been shown to have different outcomes in female patients than in males.

Response:

Thank you for your valuable suggestions. As you mentioned, ΔeGFR is known to change more slowly in women. We performed a t-test for ΔeGFR in men and women, finding no significant difference (P=0.133). The same analysis was performed for ΔTKV%, showing no significant difference (P=0.245). We have added the following sentence on page 5, line 158: " Previous studies have shown that the decrease in ΔeGFR is slower in female patients than in males [16]. A study was conducted to see if there was a difference in ΔeGFR between men and women, but the results of the t-test analysis comparing ΔeGFR in men and women showed no significant difference (P=0.133)." Additionally, on line 176 of page 6, we added: " A t-test was conducted to examine the effect of gender on ΔTKV%, but no statistically significant difference was observed between male and female patients (P=0.245)."

(2) Section 2.5: Authors should present the data for the change in ΔTKV for all 3 groups as a scatter dot plot with their significance.

Response:

Thank you for suggesting the insertion of the ΔTKV% Figure. We have produced a scatterplot of ΔTKV% and inserted it as Figure 4 on page 7.

Figure 4: Scatter dot plot of ΔTKV% at baseline, 3 years, and 6 years with HtTKV. ΔTKV% showed a significant increase in change between both baseline to 6 years (p<0.05) and 3 to 6 years (p<0.001). HtTKV, height-adjusted total kidney volume; ΔTKV%, percentage increase in TKV.

(3) Lines 154-157: "ΔTKV showed a significant increase.... at 3 years and 6 years (p<0.001)." Authors should recheck. Line 156: "increase in change both at baseline and 6 years" seems to be repeated from line 154.

Response:

Thank you very much for your helpful recommendations. The sentence was incorrect, and we have corrected it as follows (line 171): "ΔTKV showed a significant increase in change between baseline and up to 6 years (p<0.05), no significant difference in change between baseline and 3 years, but a significant increase in change 3 years to 6 years (p<0.001)."

(4) Section 2.6: Authors have analyzed multiple factors in association with the therapeutic effect of tolvaptan. However, the significance of the association should be considered after correction for multiple testing (example. Bonferroni correction) especially when various factors are analyzed simultaneously.

Response:

Thanks for your comments on this important point. The issue of multiplicity has been considered in our analysis. The model has been deemed valid with an adjusted R2 of 0.81, and all Vif values are close to 1, indicating independence. The model has been deemed valid with an adjusted R2 of 0.81, and all Vif values are close to 1, indicating independence. The Adjusted R2 and Vif values have been included in Table 3.

(5) Lines 176-177: "In addition, ΔeGFR tended.... doses of 60 gm/day or higher" implicates statistical significance between improved ΔeGFR and tolvaptan dose, which is not as per p-value of 0.06 shown in table 3.

Response:

Thank you for bringing this to our attention. The sentence has been removed. Additionally, we have included a new discussion on the dose-response relationship of tolvaptan. On page 6, Line 202, we added the following sentences: " There was no significant correlation found between tolvaptan daily doses and improvement in ΔeGFR.  A stratified analysis was performed by tolvaptan dose, with mean daily doses of less than 60mg, 60 to 90 mg, and 90 mg or more being analyzed using the Kruskal-Wallis test. However, no correlation was found between the dose and the degree of improvement in ΔeGFR in all strata (p=0.09)."

(6) Comments on the Quality of English Language Minor editing of English language required.

Response:

Your feedback is appreciated. The manuscript has been proofread by a native English speaker.

Reviewer 2 Report

Comments and Suggestions for Authors

The aim of this manuscript is to assess the impact and safety of Tolvaptan in patients with ADPKD after six years of treatment. The study includes a total of 55 cases, with baseline and 6-year post-treatment data analyzed by the author. The findings suggest that Tolvaptan treatment may slow down the progression of renal function decline. Additionally, the study highlights the significance of BMI and a family history of ADPKD as important factors influencing the effectiveness of Tolvaptan.

While the study is interesting, several modifications could enhance its overall quality, as outlined below:

  1. Performing a sub-analysis based on different Tolvaptan doses would enhance the study's quality.
  2. Clear and comprehensive figure legends are needed for Fig 2 and Fig 3.
  3. Providing a more comprehensive table legend would enhance clarity and understanding.
  4. A comparison between BMI at baseline and 6 years post-treatment could offer valuable insights.
  5. The manuscript should address why Delta TKV increases with Tolvaptan treatment, particularly in Table 2.

Author Response

(1) Performing a sub-analysis based on different Tolvaptan doses would enhance the study's quality.

Response:

Thank you very much for your invaluable comments. We have conducted a stratified analysis of the mean tolvaptan dose, which was categorized as <60 mg/day, 60-90 mg/day, and >90 mg/day. Our analysis did not reveal any significant association between the dose and renal function. We have added the results to page 6, Line 202, as follows: " There was no significant correlation found between tolvaptan daily doses and improvement in ΔeGFR.  A stratified analysis was performed by tolvaptan dose, with mean daily doses of less than 60mg, 60 to 90 mg, and 90 mg or more being analyzed using the Kruskal-Wallis test. However, no correlation was found between the dose and the degree of improvement in ΔeGFR in all strata (p=0.09)."

(2) Clear and comprehensive figure legends are needed for Fig 2 and Fig 3.

Response:

We appreciate your feedback. We have added comprehensive figure legends to Figures 2 and 3, respectively.

Figure 2: Compared to baseline, ΔeGFR at 6 years improved in 76.4% of patients, with a mean (range) change of 3.024 (-8.772-0.58) mL/min/1.73m2.

Figure 3: The effect of tolvaptan on TKV is known to decline with long-term treatment [15], and a similar trend was observed in this study.

Duplicated explanations have been omitted, and Table 1,2 itself has been revised for easier viewing.

Also, there was a typographical error in Figure 2, so we changed it as follows: Baseline ΔeGFR → 6-year ΔeGFR/year in the right figure.

In addition, the following statement is added to the legend of Table 3: "Baseline BMI and family history were statistically significant factors affecting ΔeGFR (p=0.027, p=0.044, respectively)."

(3) A comparison between BMI at baseline and 6 years post-treatment could offer valuable insights.

Response:

Thank you very much for your helpful recommendations. We conducted a study on BMI at 6 years, which revealed a mean BMI of 23.11 (range: 16.43-33.4) with a mean difference between BMI at 6 years and baseline of -0.118 (-9.27 to 7.48). We analyzed the change in BMI and improvement in ΔeGFR using Spearman's rank correlation coefficient but found no significant correlation (p=0.328). This information was added to P7 line 206 as follows: " The BMI examination at 6 years showed a mean BMI of 23.11 (range: 16.43-33.4) and a mean difference of -0.118 (-9.27-7.48) compared to the baseline. Although Spearman's rank correlation coefficient was used to analyze the change in BMI and improvement in ΔeGFR, no significant correlation was found (p=0.328)."

(4) The manuscript should address why Delta TKV increases with Tolvaptan treatment, particularly in Table 2.

Response:

Thank you for your comment. We added the following to the Table 2 legend: "The effect of tolvaptan on TKV is known to decline with long-term treatment [15], and a similar trend was observed in this study."